# Dietary Environmental Footprints and Their Association with Socioeconomic Factors and Food Purchase Practices: BRAZUCA Natal Study

**DOI:** 10.3390/foods11233842

**Published:** 2022-11-28

**Authors:** Maria Hatjiathanassiadou, Camila Valdejane Silva de Souza, Diôgo Vale, Natalie Marinho Dantas, Yasmim Bezerra Batista, Dirce Maria Lobo Marchioni, Severina Carla Vieira Cunha Lima, Clélia de Oliveira Lyra, Priscilla Moura Rolim, Larissa Mont’Alverne Jucá Seabra

**Affiliations:** 1Postgraduate Program in Nutrition, Center for Health Sciences, Federal University of Rio Grande do Norte, Natal 59078-970, RN, Brazil; 2Postgraduate Program in Public Health, Center for Health Sciences, Federal University of Rio Grande do Norte, Natal 59078-970, RN, Brazil; 3Federal Institute of Education, Science and Technology of Rio Grande do Norte, Natal 59015-300, RN, Brazil; 4Department of Nutrition, School of Public Health, University of São Paulo, São Paulo 01246-904, SP, Brazil; 5Department of Nutrition, Center for Health Sciences, Federal University of Rio Grande do Norte, Natal 59078-970, RN, Brazil

**Keywords:** sustainable diet, food consumption, environmental impact, water footprint, carbon footprint, ecological footprint

## Abstract

The analysis of dietary environmental impacts has proven to be an important tool for guiding the adoption of healthier and more sustainable diets. This study aimed to estimate the dietary carbon (CF), water (WF), and ecological (EF) footprints of residents in the city of Natal, Brazil; the study also aimed to verify their association with socioeconomic factors and food purchase practices. This is a cross-sectional study that used dietary data from 411 adults and elderlies, which was collected via a questionnaire that applied to the respondents. The results showed that the dietary CF was 1901.88 g CO_2_ eq/day/1000 kcal, the WF was 1834.03 L/day/1000 kcal, and the EF was 14.29 m^2^/day/1000 kcal. The highest environmental footprint values showed an association (*p* ≤ 0.05) with the factors of male sex, white ethnicity, and higher income and schooling, whereas the lowest environmental footprint values were associated with social vulnerability variables such as female sex, non-white ethnicity, and lower income and schooling (*p* ≤ 0.05). Moreover, people with lower environmental footprints consumed less fast food, had fewer meals at snack bars, and used food delivery services less often than those with higher footprints. The foods that most contributed to the CFs and WFs were beef and chicken, while fish and beef contribute the most to the EFs. The data in the present study show that a diet with a lower environmental impact is not always equal to a sustainable diet. This relationship is paradoxical and relates to food justice, as people with lower environmental footprint values are the same ones with worse socioeconomic conditions. In this sense, is it essential to consider the influence of the social context when assessing dietary environmental impacts and when assessing actions that promote healthier and more sustainable diets.

## 1. Introduction

Adopting healthy and sustainable dietary practices is an urgent priority in the face of current ecological and health challenges [1]. These goals require promoting dietary practices aligned with both individual needs and planetary health [2]. Such a need has become even more evident in the scientific and political fields after the publication that pointed out the present global syndemic, i.e., the co-existence of undernutrition, obesity, and climate change pandemics around the world [3].

The implementation of more sustainable actions has become an international agenda after the agreement with respect to the Sustainable Development Goals (SDGs) by member states of the United Nations. The incentive to establish more sustainable food production and consumption standards are included in the scope of actions for the promotion of global health, particularly in SDGs 2, 11, 12, 13, 14, and 15 [4]. The strategies to reach those goals begin with the comprehension of dietary indicators in populations and their environmental impacts. Some studies employ environmental sustainability metrics linked to food systems, such as food waste [5] and the estimated environmental footprints (EnF) of individual food consumption [6].

In the field of nutrition, studies using ecological (EF) [7,8,9], water (WF) [10,11,12,13,14], and carbon (CF) [15,16,17,18] footprints to assess the dietary sustainability of population groups have become frequent. These indicators are interesting as they enable laying the basis for educational action and public policies aimed at reducing environmental impacts and improving health through diet [19,20]. EnF estimates in different regions point to differences according to the dietary standards of the territories. In South America, a study performed in Argentina showed values of 8910 g CO_2_ eq/day for greenhouse gas (GHG) emissions, 54.2 m^2^/day for soil use, and 205 L/day for freshwater [21], while a study performed in Chile showed GHG and soil use values of 4670 g CO_2_ eq/person/day and 4177 L/person/day, respectively [22]. In Asia, a study performed in Lebanon determined the water use of food consumption to be 2571 L/day and the GHG emissions to be 4060 g CO_2_ eq/day [23]. In some Mediterranean cities of Europe and Asia, the WF values of food consumption have been shown to range from 3272 L/per capita/day to 5789 L/per capita/day [24].

In Brazil, a study with data on household food availability showed an increase by 21%, 22%, and 17% in the CF, WF, and EF, respectively, between 1987 and 2018 [9]. Another study analyzed the EnF of food consumption data of Brazilians and identified averages of 6.76 kg CO_2_ eq/person/day for CF, 3478.4 L/person/day for WF, and 67.2 m^2^/person/day for EF [25]. It is noteworthy that Brazil is the fourth-highest contributing global economy to GHG emissions from the food system, using large amounts of soil and water for food production [26]. In this context, the importance of the advancing analyses in Brazil stands out, as many of the current estimates focus on the national level [9,25,27,28]. These estimates need to be expanded to the regional level as diets are quite varied in the country, which has continental dimensions and one of the largest populations in the world.

Brazil has made advancements in promoting sustainable diets. One of the examples is the Dietary Guidelines for the Brazilian Population, which was one of the first guidelines in the world to discuss social, cultural, economic, and other aspects of sustainability. Along with these inclusions, the document discusses the impacts of food processing by using the NOVA classification system to base the main recommendations, which stimulates the development of more sustainable and healthy diets by recommending a diet based on in natura and minimally processed foods of plant origin, with one of its principles being the sustainability of dietary systems [9,29,30].

In this sense, the present study sought to assess the dietary environmental impacts in the city of Natal, the capital of the state of Rio Grande do Norte (RN), Brazil. Natal is a coastal city with approximately 896,708 inhabitants and is the most populous municipality in the state, the 6th most populous in the Northeast region, and the 20th in Brazil. The city has a Municipal Human Development Index (MHDI) of 0.763, which is considered high [31,32]. In Natal, foods such as fish and seafood, particularly shrimp, and products from the small- and medium-sized cities of the state such as sun-dried meat and cassava flour are characteristics of the city’s cuisine [33]. On the other hand, RN is the state in the Northeast region of Brazil that consumes ultra-processed foods the most and the one that consumes in natura or minimally processed foods the least in the North and Northeast regions [34].

The importance of this study is justified through the need to further studies on dietary impacts given the urgent priority associated with the matter and the need to assess the diet of different population groups according to their social, economic, and cultural contexts. Such studies are key to promoting health actions at the national and regional levels, thus fostering a healthier and more sustainable diet for all. Therefore, the present study sought to answer the following questions: (1) What are the values of the environmental footprints of food consumption for adult and elderly residents in Natal? (2) Is there a relation between the environmental footprints of food consumption and the socioeconomic characteristics and food purchase practices?

## 2. Material and Methods

### 2.1. Study Characterization

This is an applied field research with a cross-sectional, observational, and quantitative design. The data in the present study come from a research project entitled “Food Insecurity, Health and Nutrition Conditions in an Adult and Elderly Population of a Capital in the Northeast Region of Brazil: BRAZUCA Natal/RN Study” (Insegurança Alimentar, Condições de Saúde e de Nutrição em População Adulta e Idosa de uma Capital do Nordeste do Brasil: Estudo BRAZUCA Natal/RN), which is part of a population-based multicenter survey called the Brazilian Usual Consumption Assessment (Estudo BRAZUCA).

A two-stage (census sectors and households) probabilistic sampling was performed. Sixty-six effective census sectors and six alternate census sectors were randomly chosen. The number of households was defined considering the minimum sample size, and the density of elements of each demographic group per household was calculated from data from the 2010 Brazilian Census with a 10% correction rate to account for losses from refusals and closed and vacant households. The final sample size aimed to reach 258 interviews in each of the four sex and age strata: adults and elderly of either sex (total of 1032 people). The minimum size of 258 in each stratum allowed for us to estimate a prevalence of 50% with an 8% error and 95% confidence level. The design effect (deff) factor was 1.5, and 15% were added as a rate of non-responses and closed households.

This study presents a cropping of data collected from June 2019 to March 2020 in 27 census sectors in the four sanitary districts of the municipality of Natal, RN. Adult and elderly persons (≥20 years) of either sex who were physically and cognitively able to answer the questionnaires were considered eligible for the research. The present study considered all interviews conducted during the collection period, with a total of 411 persons being characterized as a convenience sample.

### 2.2. Ethical Aspects

The BRAZUCA Natal/RN Study was submitted to and approved by the Committee of Ethics and Research of the University Hospital Onofre Lopes (CAAE 96294718.4.2001.5292), under Protocol No. 3531,721. The Estudo BRAZUCA application was also approved (CAAE 96294718.4.1001.5421). The study was conducted in accordance with the Declaration of Helsinki and Resolution No. 466 of 12 December 2012 of the National Health Council [35]. The individuals eligible to take part in the research were informed about the objectives, risks, and benefits, and those who accepted to participate signed a term of free and informed consent.

### 2.3. Data Collection and Instruments Used

The data were collected in households or at a primary healthcare unit close to the residence of participants. The interviews were performed using a standardized and revised questionnaire based on the protocols of the National Health Survey (Pesquisa Nacional de Saúde—PNS 2013) applied via the Epicollect5 platform. All steps were guided by manuals and standard operating procedures (SOPs) developed by professors and Ph.D. students. All interviewers were properly trained and qualified. The interview collected information regarding the demographic and socioeconomic data, food consumption, and food purchase practices.

Food consumption data were obtained using a propensity questionnaire (PQ). The questionnaire aimed to analyze the intake of 41 food groups over the 12 months prior to the interview. For each group, participants were asked to indicate the frequency of consumption (in days, weeks, months, or years) and the number of times (from 1 to 10). The questionnaire did not assess portion intake or amounts. The estimates of per capita amounts for each individual were taken from the information collected in a 24-h Dietary Recall (24 HR), which was applied on the same day as the interview using the GloboDiet software [36].

### 2.4. Study Variables

#### 2.4.1. Hypotheses

Based on the research questions “What is the value of the environmental footprints of the food of adults and elderly residents in the city of Natal/RN?” and “Is there a relationship between the environmental footprints of food consumption and socioeconomic characteristics and food purchasing practices?”, the following hypotheses were elaborated (Table 1).

#### 2.4.2. Socioeconomic Characteristics

Information was gathered on the biological sex (female and male), age group (adults 20–59 years; elderly ≥60 years), ethnicity (white and non-white: black, yellow, pardo (official term used by the IBGE census referring to people of mixed race), indigenous), schooling (0–5 years: elementary school; 6–9 years: middle school; 10–13 years: high school; and ≥14 years: higher education), and monthly per capita family income (BRL), which was split into quintiles: <249.50 (Q1); 249.50–449.45 (Q2); 449.46–762.67 (Q3); 762.68–1559.99 (Q4); ≥1560.00 (Q5).

#### 2.4.3. Assessment of Food Purchase Practices

The information regarding food purchase frequency refers to purchases from street fairs, fast food restaurants, and the use of food delivery services, which was split into usage categories of “never”, “sometimes”, and “often”. The frequency of having lunch or dinner at snack bars was expressed as “never”, “hardly ever”, and “at least once a week” (the original response options of “1–2 days a week”, “3–4 days a week”, “5–6 days a week’ and “every day” have been condensed into “at least once a week”).

#### 2.4.4. Estimated Environmental Footprints of Food Consumption Frequency

The footprints of food consumption frequency were estimated using the PQ. As the PQ did not assess portion intake or amounts, these data were estimated based on the per capita values from the 24 HR. For each food group, the median of the number of times it was consumed was estimated, and a weighted average was calculated. Some foods need to be diluted, as the same food was estimated in milliliters (mL) and/or in grams (g), depending on how each individual reported consumption. This was the case with powdered milk and chocolate powder. For dilution, we used the rules according to Araújo and Guerra [39]. On the other hand, foods consumed in liquid form (mL or L) such as coffee and tea needed to be transformed into kilograms, as this is the unit of measurement used to analyze the EnF. Thus, these were also converted following the same dilution rules (Appendix A).

After estimating the per capita amount of each food group, the WF, CF, and EF values were estimated. For this calculation, the frequency of daily consumption was considered for each food group from the PQ. To that end, the consumption frequency data collected in years, months, or weeks were converted into daily consumption: the values estimated in a year were divided by 365; those estimated by month were divided by 30; and those estimated by week were divided by 7.

After the daily consumption frequency was obtained, the values were multiplied by the per capita amount to obtain the estimated food consumption (EFC), as described in Equation (1) [28,40]:(1)$$EFC=daily\;consumption\;frequency\times \left(\frac{per\;capita\;\left(g\right)}{1000}\right)$$

The per capita consumption value was converted from grams into kilograms to correct the unit of measurement. The final value was expressed as kilograms per day (kg/day).

To estimate the mean environmental footprint value of each food for each person, the EFC (Equation (1)) was multiplied by the environmental footprint values for foods and preparations consumed in Brazil, as described by Garzillo et al. [40], which takes into account the whole lifecycle (from farm to fork), using the life cycle assessment (LCA) methodology. Therefore, the values of prepared foods consider correction factors, cooking factors, and estimated carbon emissions associated with the different types of preparation.

The footprint values of raw and prepared foods were considered, and the weighted average was performed considering the EnF values and number of times the food was consumed according to what was observed in the 24 HR. Adaptations were made when foods reported in the 24 HR were not present in the footprint database (Appendix A). For example: For the fruit group, all fruit consumed by the participants according to the 24 HR and their respective number of times and way of consumption/preparation were compiled. After the compilation, the weighted average was calculated for each footprint, yielding the mean values of CF, WF, and EF for the fruit group.

To obtain the final footprint of each individual, the environmental footprint values calculated from all food groups consumed were added up, as shown in Equation (2) [23,28]:(2)$$Final\;footprint\;of\;the\;individual=\overset{Food\;groups\;consumed}{\underset{i=1}{\sum }}EFC_{i}\times impact_{i}\;$$
where i is each food group consumed and “Food groups consumed” is the total amount of groups present in the PQ, considering the consumption informed by the individual. The EFC_i_ refers to the EFC of each food group (Equation (1)), while impact refers to the environmental footprint values for each group according to the type of footprint. This formula was used to calculate the CF, which was measured in grams of CO_2_ equivalent (g CO_2_ eq), the WF, which was measured in liters (L), and the EF, which was measured in square meters (m^2^).

After the estimated environmental footprint of each individual was calculated (Equation (2)), the values were equalized to 1000 kcal per person. To this end, the total caloric value was calculated considering the estimated daily consumption (Equation (1)) and the caloric estimated from the PQ. The caloric estimative from the PQ was calculated considering the EFC value in grams of each food group and the caloric value from the Brazilian Table of Food Composition (Tabela Brasileira de Composição de Alimentos—TBCA) [41] considering a 100 g portion. For each food group, the arithmetic mean was calculated considering all the foods present in the TBCA that fit the respective group. The calculation is explained in Equation (3) [23,28]:(3)$$Ekal=\overset{Food\;groups\;consumed}{\underset{i=1}{\sum }}EFC_{i}\times Kcal_{i}\;$$
where i is each food group consumed and “Food groups consumed” is the total amount of groups present in the PQ, considering the consumption informed by the individual. The EFC_i_ refers to the EFC of each food group (Equation (1)), while Kcal_i_ refers to the average value of kilocalories per 100 g of food in each group.

With the per capita caloric consumption estimated (Equation (3)), the environmental footprint values were equalized for each 1000 kcal per capita per day, as described in Equation (4) below [23,28]:(4)$$EEF=\frac{Final\;footprint\;of\;the\;individual\times 1000\;kcal\;}{EKcal}$$

The final values can be expressed as g CO_2_/person/day/1000 kcal for CF, L/person/day/1000 kcal for WF, and m^2^/person/day/1000 kcal for EF.

### 2.5. Statistical Analyses

The main variables used were the estimated dietary environmental footprints (CF, WF, and EF) adjusted to 1000 kcal. The secondary variables were the socioeconomic characteristics and assessment of food purchase practices.

The results were expressed as the median, mean, interquartile interval, maximum and minimum values, and CI 95% frequency. Data normality was assessed using the Kolmogorov–Smirnov test. Because the data presents a non-normal distribution (*p* < 0.05), non-parametric tests (Mann–Whitney U test and Kruskal–Wallis test) were used to assess the differences between the EnF medians. The missing data answered as “doesn’t know or didn’t answer” by the participants were ignored in the data analysis and were indicated in the respective tables and figures.

The environmental footprint values were also assessed, taking tertiles into account; food contribution per tertile was analyzed, with the values expressed as percentages. The correspondence analysis (CA) test was performed to verify the association between the environmental footprints (main variables) and the secondary variables. The first, second, and third tertiles were identified as low, medium, and high footprints, respectively.

The CA plots show the graphical representation of the associations, where the tertiles are represented by ellipses and the other variables are represented by points. An association exists when those points are within one or more ellipses. The association is considered significant when the *p*-value is below 0.05 and when the chi-squared value observed is lower than the critical chi-squared.

The CA test is an exploratory statistical technique that enables the graphical visualization of the relationships of a large set of variables amongst themselves. In this sense, the technique allows for the identification of associations or similarities between the qualitative variables or categorized continuous variables. The relationship among variables is seen with no need to assign a causal relationship and without assuming a distribution of likelihoods, which makes it appropriate for use in populational studies [42,43,44,45].

The data were tabulated in the software Microsoft Excel^®^ and analyzed using IBM^®^ SPSS^®^ Statistics and XLSTAT software. The statistical significance was defined as *p*-value ≤ 0.05. Figure 1 presents an overall flowchart of methodology.

## 3. Results and Discussion

### 3.1. Estimated Dietary Environmental Footprints

Table 2 presents the estimated dietary environmental footprints of adults and elderly people participating in the study. The total values of 2678.53 g CO_2_ eq/day for CF, 2702.53 L/day for WF, and 20.47 m^2^/day for EF were found. With the adjustment to 1000 kcal, the values are 1901.88 g CO_2_ eq/day/1000 kcal for CF, 1834.03 L/day/1000 kcal for WF, and 14.28 m^2^/day/1000 kcal for EF.

We compared the results of this study with previous studies performed in Brazil and in other countries and regions that have standardized environmental footprints to 1000 kcal. Some studies presented values similar to ours. This is evident in the case of a Canadian study where the dietary CF was estimated at 2150 g CO_2_ eq/day/1000 kcal [17] and a study performed in the United States [18] where the CF was 2210 g CO_2_ eq/per capita/day/1000 kcal. Some studies presented higher values; in a study performed in Sweden, the dietary CF in 2016 was 3380 g CO_2_ eq/1000 kcal/day [46]. Some studies presented lower values, as in a study conducted in Lebanon, where the WF was 951.68 L/day/1000 kcal and CF of 1530 g CO_2_ eq/day/1000 kcal in adult diets [23]. Another study also conducted in Lebanon assessed the consumption of the Mediterranean diet by Lebanese adults and reported WF of 995.79 L/day/1000 kcal and CF of 0.68 kg CO_2_ eq/day/1.000 kcal [47].

In Brazil, according to household food purchase data from 2017–2018, the CF was 1866 g CO_2_ eq/day/1000 kcal, the WF was 1769 L/day/1000 kcal, and the EF was 11.36 m^2^/1000 kcal [9]. These values were close to those found in our study.

Estimates at the national and regional levels are equally important for understanding the environmental impacts associated with diets, given the difference in eating habits between populations. In the Brazilian context, analyses at the regional level become even more important, as habits differ according to region and location. Brazil is a country of continental dimensions, with a land area of approximately 8.5 million km^2^, close to the size of the European continent (10.2 km^2^) [48,49]. In this sense, diets heavily vary in Brazil as these habits were and are influenced by several factors, such as social context and historical factors. Understanding how different diets relate to environmental impact is essential for guiding actions aimed at mitigating the advance of climate change. It is also important to highlight that the differences in EnF values between the studies may be associated with the eating habits of each location, source of the footprint values used, and system boundaries according to the LCA methodology.

Given the division into tertiles, the median values were also estimated for each footprint. The low CF group had a median of 1372.99 gCO_2_eq/1000 kcal; the medium CF group had a median of 1.901,88 gCO_2_eq/1000 kcal; and the high CF group had a median of 2777.67 gCO_2_eq/1000 kcal. The low WF group had a median of 1436.58 L/1000 kcal; the medium WF group had a median of 1834.03 L/1000 kcal; and the high WF group had a median of 2430.40 L/1000 kcal. Finally, the low EF group had a median of 9.80 m^2^/1000 kcal; the medium EF group had a median of 14.29 m^2^/1000 kcal; and the high EF group had a median of 21.01 m^2^/1000 kcal.

### 3.2. Environmental Footprints and Populatin Characteristics

Table 3 shows the characteristics of the population studied and their respective environmental footprints.

Regarding socioeconomic characteristics, adults (*p* = 0.00), male individuals (*p* = 0.05), and monthly per capita family income of BRL 1560.00 or more (*p* = 0.00) had higher CF values. For WF, male individuals (*p* = 0.02), adults (*p* = 0.01), and monthly per capita family income of BRL ≥ 1560.00 (*p* = 0.00) had the highest values. No statistically significant differences were seen for EF.

Such results match the findings of other studies. In Brazil, males and adults had the highest contributions to CF, WF, and EF in the study by Travassos, Cunha, and Coelho [25], as well as to CF in the study by Garzillo et al. [28]. In Sweden [46], India [14], China [50], and the United States [18], males also had the highest contributions to dietary CF and WF values.

Garzillo et al. [28] also found that the CF of the Brazilian diet increase with income and education. In the study by Song et al. [50], Chinese family income had a strong impact on the consumption of foods of animal origin, which are the foods that have the highest EnF, and less impact on the consumption of foods of plant origin. In this sense, it is worth pointing out the difference in dietary EnF analysis in developed and developing countries. The lower environmental footprints in developing countries such as Brazil would be associated with the low purchasing power of the population and consequent lower access to foods with higher environmental impact, such as foods of animal origin [28,51,52]. We emphasize that 64.5% of the population in this study is below the poverty line proposed by the World Bank [53], i.e., they survive with less than USD 5.50 per day. This scenario refers to the conditions before the COVID-19 pandemic and has worsened after the pandemic. Data from 2021 and 2022 show that 125.2 million Brazilian households were in a food insecurity situation, while 33 million were in severe food insecurity [54].

It is also worth highlighting the difference in protein sources among the groups presented in Table 3. We observed a higher consumption of fish by the groups with lower per capita family income, lower schooling, and by seniors. Such a difference in fish consumption may have impacted the EF values, explaining the higher EF value observed, for example, in the group with up to five years of schooling. The high fish consumption and its impact on EF values were observed in a previous study performed in Natal [55], which points out the high consumption of cheaper meat cuts, processed meats, and higher consumption of fish and seafood by Natal residents when compared with the average Brazilian, which would increase the demand for fishing grounds, one of the ecological resources considered in the EF analysis [55].

Studies in other countries also show the impact of fish consumption on EF values. In Portugal, it was observed that the high consumption of animal protein, particularly fish and red meat, negatively impacted the EF. The consumption of fish and seafood contributed to approximately 26% of the total EF in Portugal, which is even higher than the consumption of meats (23%) [56].

Regarding food purchase practices, those who bought fast food “sometimes” or “often” had higher CFs (*p* = 0.00) and WFs (*p* = 0.00) than those who never did. It was also observed that those who had lunch or dinner at snack bars at least one day a week had higher CFs (*p* = 0.00) and WFs (*p* = 0.02). No statistical difference was observed in the environmental footprint results considering the frequency of food purchases at street fairs. Individuals who never used food delivery services had lower CF values when compared with those who used them often (*p* = 0.03).

We point out that the higher CF and WF values associated with those practices may be related to the higher consumption of ultra-processed foods and meat. A Brazilian study reported that 70% of all food supplied in food delivery applications are ultra-processed or ready-to-eat preparations [57]. Another study stated that the most often-bought foods at Brazilian snack bars were savory snacks and fast food [58]. Some studies have explored the relationship between the higher consumption of ultra-processed foods and their impacts on EnF values. A study found that ultra-processed foods significantly contributed to the increase in dietary CFs, WFs, and EFs of Brazilians between 1987 and 2018, accounting for an increase by 245% in CFs, 233% in WFs, and 183% in EFs over the years, whereas no statistically significant difference was found for the contribution of in natura or minimally processed food during that period [9]. The study by Vale et al. [37] reported a progressive increase in WF values in the diets of Brazilian adolescents as the number of days they ate at fast food restaurants increased, which are businesses that normally sell ultra-processed foods.

It is noteworthy that despite the increase in the consumption of ultra-processed foods being associated with higher EnF values, the profile of the ultra-processed foods consumed is important. A larger amount of meat products within the group of ultra-processed foods contributes to the increase in EnF values [27]. In addition, we also point out that the analysis of environmental impacts still requires further research. The current analyses based on LCA, such as footprints, does not consider the industrial process or the use of several compounds, such as chemical additives, and the use of large amounts of packaging, which may mask an even higher impact [59].

### 3.3. Association among Environmental Footprints, Socioeconomic Characteristics, and Food Purchase Practices

Figure 2 shows the graphical representation of the associations of EnF tertiles (ellipses) with the remaining variables (dots). The association was considered significant (*p* < 0.05) (Appendix A) for CF and WF (Figure 2 and Figure 3).

The variables associated (*p* = 0.00) with the lowest CF values (Low CF) (Figure 2) were non-white, having zero to five years of schooling, having a monthly per capita family income of up to BRL 249.50, never buying fast food, never using food delivery services at home, never buying food at street fairs, and never having lunch or dinner at snack bars. The populational values associated with the highest CF values (High CF) were male, adult, white, having 14 or more years of schooling, monthly per capita family income equal to or above BRL 1559.99, sometimes buying fast food, and sometimes or often using food delivery services at home.

The variables associated (*p* = 0.00) with the lowest WF values (Low CF) (Figure 3) were female, elderly, non-white, having zero to five years of schooling, never buying food at fast food restaurants, never using food delivery services at home, and never having lunch or dinner at snack bars. The highest WF values (High WF) were associated with male, adult, white, having 14 or more years of schooling, sometimes or often buying fast food, sometimes or often using food delivery services at home, and having lunch or dinner at snack bars at least once a week. For EF values (Figure 4), no significant association was observed (*p* = 0.21).

It was found that the lowest CF and WF values were associated with values related to social vulnerability, such as being female, non-white, having a lower schooling, and having a lower per capita family income. Meanwhile, the highest CF and WF are associated with variables related to better living conditions, such as being male, white, having a higher schooling, and having a higher per capita family income.

The highest footprints were also associated with a greater frequency of purchasing food at fast food restaurants and snack bars, more frequent use of food delivery services, and lower frequency of food purchases at street fairs. In this sense, although the variables of the groups with high footprint are associated with better quality of life, that does not necessarily suggest good dietary quality.

Figure 5 presents the foods that most contributed to the value of the three footprints analyzed.

Foods of animal origin had the largest contribution to all three footprints, especially meats, particularly beef, chicken, and fish. Red meat stood out in CF and WF, contributing 48% and 37% to the data, respectively, considering both in natura and jerked beef. For EF, fish stood out with a 26% contribution, followed by beef with an 18% contribution. Several studies have also reported the impact of meat consumption, especially beef, on dietary EnF values, as well as the impacts associated with the production of these foods [1,3,16,18,25,60,61]. In Brazil, meat production, in addition to being associated with environmental impacts such as deforestation and GHG emissions, was also associated with slave labor situations. The livestock sector had the highest number of such cases between 1995 and 2020, accounting for 51% of notified cases over that period [62].

Figure 6 shows the foods that most contributed to the environmental footprints, which were divided into low-, medium-, and high-footprint groups.

Meats were the main contributors to the footprint values, especially in the tertile concerning the highest EnF values. For CF, the meat group accounted for 88.0% of the total value in the group with the highest CF, whereas its value was around 63.1% in the group with the lowest CF. For WF and EF, the values were 73.4% and 79.3% in the group with the highest footprint and 50.3% and 60.80% in the group with the lowest footprint, respectively. Although the higher contribution of the meat group, especially red meat, is seen in the third tertile (high footprint values), meats also had a sizeable contribution to EnF values in all tertiles, differing only in the type. Participants in the low-footprint group, for example, consumed more chicken and fish. This observation can be interpreted as a possible replacement of red meat in the face of difficulties in its purchase, which is mainly associated with affordability.

Furthermore, we observed that individuals in the low-footprint group consumed more bread, sugar, couscous, and tapioca, which are more affordable foods. In turn, those in the high-footprint group consumed more dairy products (milk, cheese, and yogurt) when compared with the low-footprint group.

The higher consumption of affordable foods such as sugar and bread, associated with lower consumption of dairy and meats may be related to the socioeconomic characteristics associated with the group and its consequent influence on purchasing more expensive foods. Such a result was also found in previous studies [17,18,25,46].

The difficulty in purchasing more expensive foods, such as those of animal origin, and the higher consumption of more affordable foods also is reflected in food purchase patterns. Those with high EnF values were the ones who bought more fast food, ate more often at snack bars, and used food delivery services more often when compared with individuals with lower EnF values. The 2017–2018 Household Budget Survey (Pesquisa de Orçamentos Familiares—POF) showed that the higher the income, the greater the expenses of eating away from home [63].

Several studies around the world have also explored the hypothesis of the influence of individual characteristics on food purchases and the consequent impacts on footprint values. In China, individuals in the upper classes had a higher environmental impact due to the consumption of foods of animal origin when compared with those in the lower classes. Meat consumption has also been considered a major contributor to the increase in CF, WF, and land use [64]. In the United States, individuals with better socioeconomic status, which accounted for greater environmental impacts (GHG emissions, water use, land use, and energy consumption) and was especially related to the higher consumption of livestock foods, such as meats, milk, and dairy products [65]. In a Swedish study, the authors argued that higher schooling, associated with higher income, enables more frequent consumption of foods that contribute to footprint values such as meat and dairy [46]. In Brazil, it was found that the CF increased along with income and schooling [28].

It is, therefore, important to discuss not only the need to change dietary standards but also the widespread access to quality diets. The study by He et al. [65] reported that 38% of Blacks and Hispanics in lower classes with lower schooling were unable to adhere to a healthy and sustainable dietary standard, as such diets can cost more and would become a hindrance to behavior change. In this sense, non-white female individuals with a lower family income and lower schooling would possibly require greater effort to change diets than white males with a higher family income and higher schooling.

Nevertheless, the relationship can be paradoxical. Access to quality diets can result in an increased consumption of food of animal origin, thus increasing the environmental impact. In this regard, it is necessary to work on actions at the individual and population levels. Scientific research is essential to work on change; however, as described by Willet et al. [1], the full range of policy levers is likely required. It can start initially with soft policy interventions, such as consumer advice, information, education, and labeling. However, it is not possible to expect that the whole food system and the impacts associated with it will be modified by only soft policy interventions. More incisive actions that involve all stakeholders are necessary so that real change can be observed, such as laws, fiscal measures, subsidies, and penalties.

In view of this, it is important to highlight the need to restructure the food system at all levels by changing the way foods have been produced, rethinking agricultural and livestock models, attributing responsibilities to the food and beverage industry, and revising consumer behavior, which are seen as aspects that directly relate to environmental, social, economic, health, and cultural issues.

### 3.4. Limitations and Strengths of the Study

As limitations, we highlight that the EnF database employed, despite considering the different forms of preparation, was based on a review of published studies, which may not precisely reflect the footprint values of the population in the present study, as many EnF estimates were performed in other countries. We also emphasize that the size sample may not be adequate to draw a profile of the environmental impact of the food consumption of people residing in Natal/RN. For this reason, some statistical analyses could not be performed due to the non-normal distribution, which may have compromised the strength and precision of the analyses.

One major strength is the originality of the research. As far as we know, there are no studies on the dietary environmental impact and its association with socioeconomic characteristics and food purchase practices of adults and elderly people in the Northeast region of Brazil. The importance of the results herein presented is also noteworthy, as they can be used to guide future studies and public policies towards the promotion of behavior change among the population for the purpose of healthier and more sustainable diets.

## 4. Conclusions

This study estimated the EnF of adults and elderlies living in the city of Natal, RN, Brazil, and investigated its association with socioeconomic characteristics and food consumption and purchase practices. We conclude that individual characteristics such as sex, ethnicity, schooling, and per capita family income directly impact access to foods and, consequently, the environmental footprint values. We also point out that food purchase behaviors, such as a greater frequency of the use of food delivery services, consumption of fast food, and having meals at snack bars are also associated with higher CF and WF values. Therefore, it is important to emphasize that lower footprint values do not necessarily indicate a healthier and more sustainable diet. In this sense, it is important to develop a broader outlook over issues involving diet, which are seen as dietary choices that are influenced by the social and economic context of individuals. Studies with a larger sample, intervention studies, and government actions on the food system are necessary to move towards more sustainable and healthy diets.

## Figures and Tables

**Figure 1 foods-11-03842-f001:**
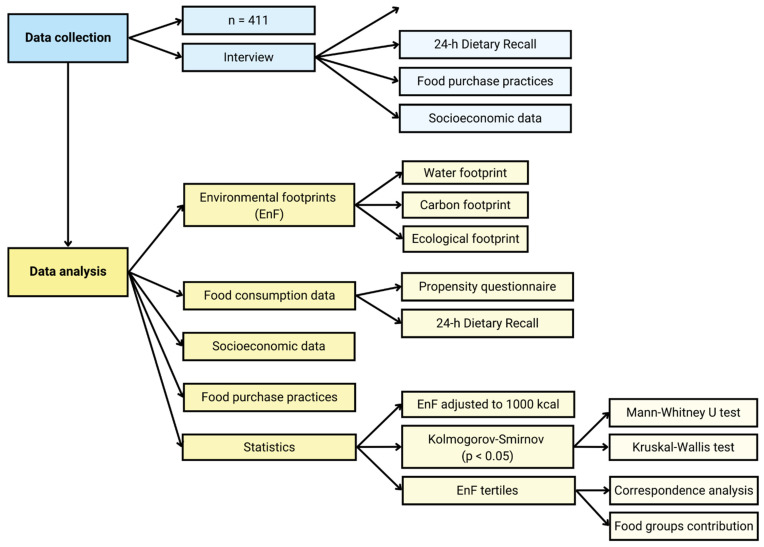
Flowchart of the methodology.

**Figure 2 foods-11-03842-f002:**
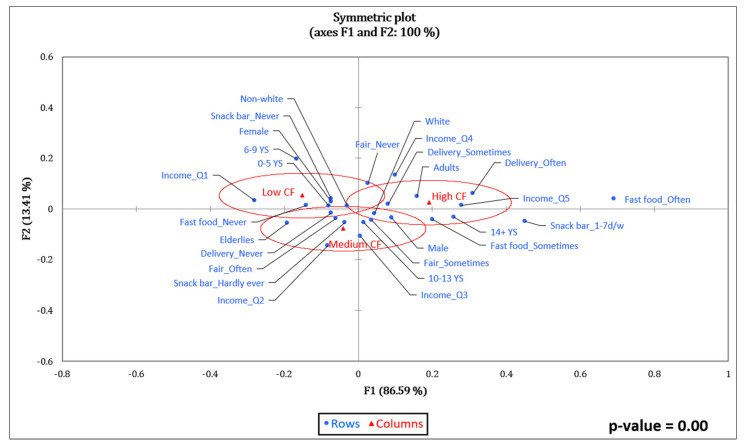
Correspondence analysis between carbon footprint and socioeconomic variables and food purchase practices of adults and elderly people taking part in the BRAZUCA Natal study (*n* = 411). 0–5 YS = 0–5 years of schooling; 6–9 YS = 6–9 years of schooling; 10–13 YS = 10–13 years of schooling; 14+ YS = 14 or more years of schooling; Income_Q1 ≤ BRL 249.50; Income_Q2 = BRL 249.50–449.45; Income_Q3 = BRL 449.46–762.67; Income_Q4 = BRL 762.68–1559.99; Income_Q5 ≥ BRL 1560.00.

**Figure 3 foods-11-03842-f003:**
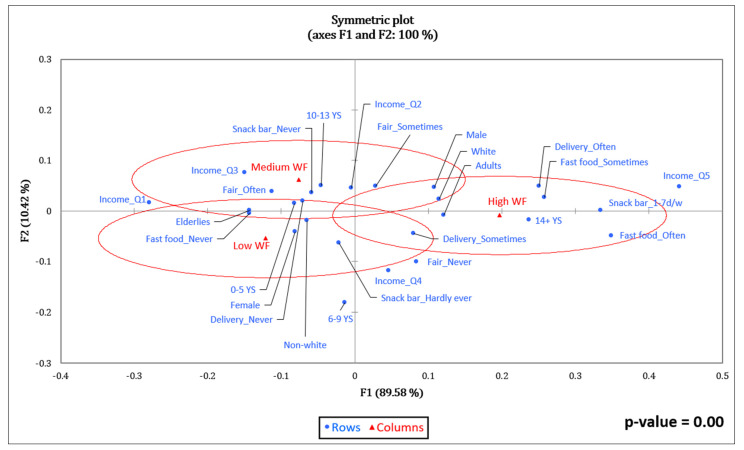
Correspondence analysis between the water footprint and socioeconomic variables and food purchase practices of adults and elderly people taking part in the BRAZUCA Natal study (*n* = 411). 0–5 YS = 0–5 years of schooling; 6–9 YS = 6–9 years of schooling; 10–13 YS = 10–13 years of schooling; 14+ YS = 14 or more years of schooling; Income_Q1 ≤ BRL 249.50; Income_Q2 = BRL 249.50–449.45; Income_Q3 = BRL 449.46–762.67; Income_Q4 = BRL 762.68–1559.99; Income_Q5 ≥ BRL 1560.00.

**Figure 4 foods-11-03842-f004:**
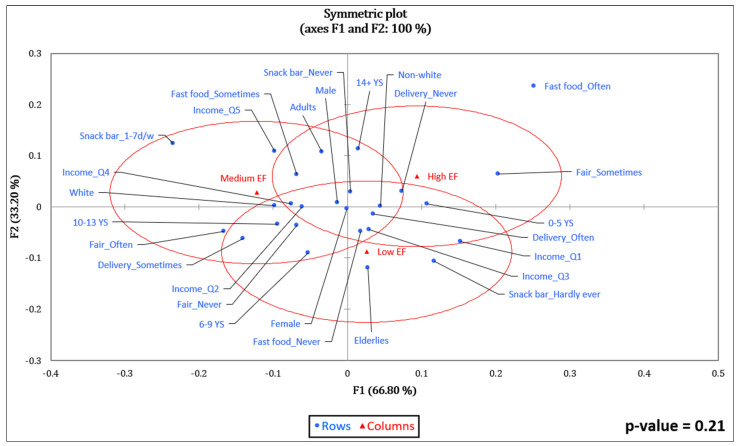
Correspondence analysis between the ecological footprint and socioeconomic variables and food purchase practices of adults and elderly persons taking part in the BRAZUCA Natal study (*n* = 411). 0–5 YS = 0–5 years of schooling; 6–9 YS = 6–9 years of schooling; 10–13 YS = 10–13 years of schooling; 14+ YS = 14 or more years of schooling; Income_Q1 ≤ BRL 249.50; Income_Q2 = BRL 249.50–449.45; Income_Q3 = BRL 449.46–762.67; Income_Q4 = BRL 762.68–1559.99; Income_Q5 ≥ BRL 1560.00.

**Figure 5 foods-11-03842-f005:**
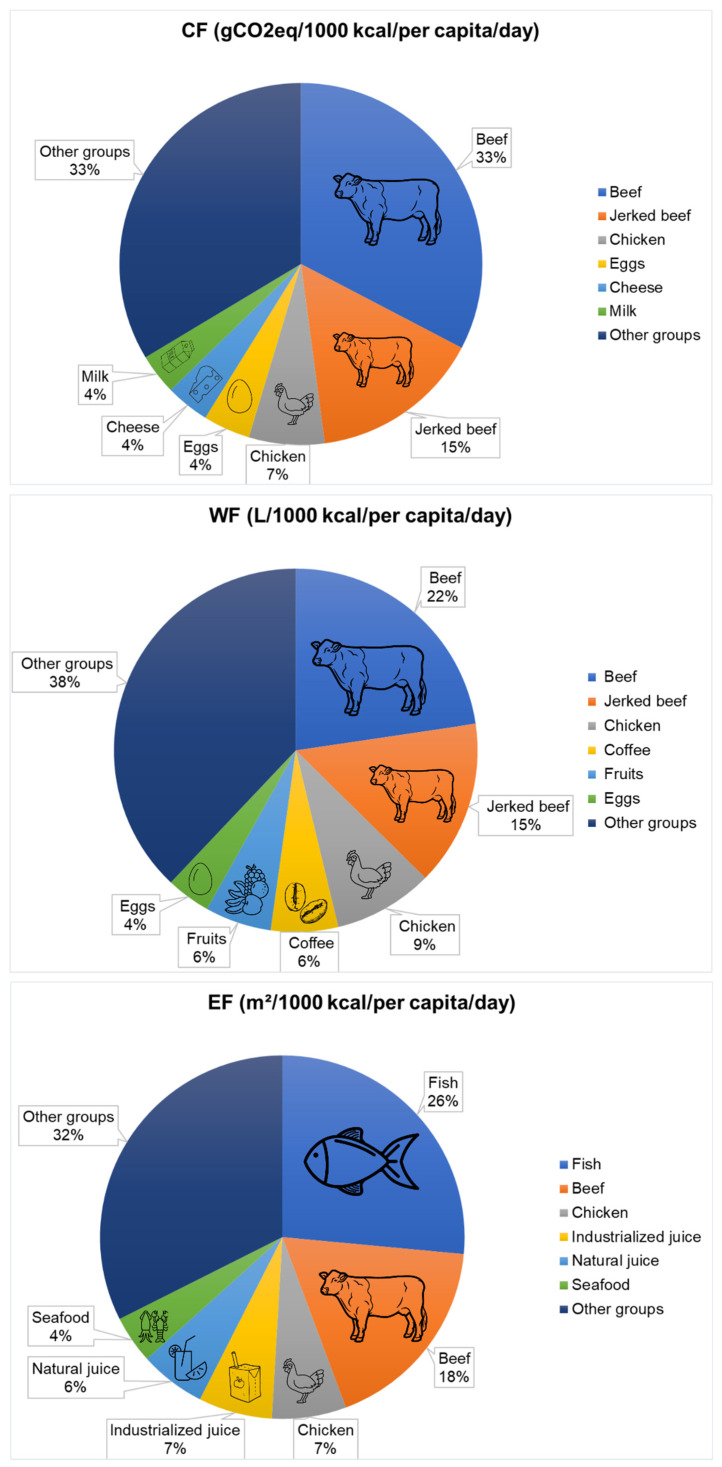
Contribution of foods to the values of the carbon, water, and ecological footprints based on the answers to the propensity questionnaire by the participants of the BRAZUCA Natal study (*n* = 411).

**Figure 6 foods-11-03842-f006:**
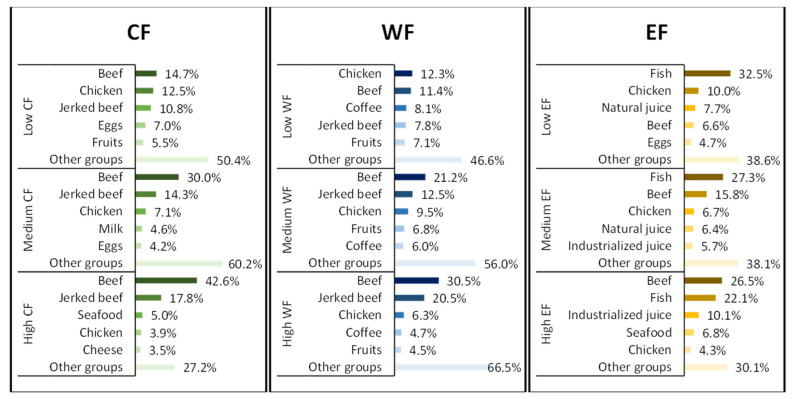
Contribution of foods to the values of carbon, water, and ecological footprints split into low, medium, and high footprints based on the answers to the propensity questionnaire by the participants of the BRAZUCA Natal study (*n* = 411).

**Table 1 foods-11-03842-t001:** Hypotheses.

Hypotheses	Metrics	References
Socioeconomic characteristics influence the dietary environmental footprints of adults and elderly people living in Natal, RN, Brazil.	Dietary environmental footprints (CF estimated in gCO_2_/person/day/1000 kcal, WF estimated in L/person/day/1000 kcal, EF estimated in m^2^/person/day/1000 kcal) and their relationship with socioeconomic variables (sex, age, ethnicity, schooling, and monthly per capita family income).	[3,14,18,19,25,28,37,38]
Food purchase and consumption practices influence the dietary environmental footprints of adults and elderly living in Natal, RN, Brazil.	Dietary environmental footprints (CF estimated in gCO_2_/person/day/1000 kcal, WF estimated in L/person/day/1000 kcal, EF estimated in m^2^/person/day/1000 kcal) and their relationship with food purchase practices (food purchase frequency from street fairs, fast food restaurants, use of food delivery services and food consumption at snack bars).	

**Table 2 foods-11-03842-t002:** Estimated environmental footprints (water, carbon, and ecological) of daily per capita food consumption of adults and elderly people living in the municipality of Natal/RN, Brazil, BRAZUCA Study.

Variables	Median	IQR	Min–Max
**CF (gCO_2_eq/person/day**)			
Estimated footprint ^1^	2678.53	1970.70–3636.07	462.13–17,066.11
Adjusted to 1000 kcal	1901.88	1493.11–2503.21	830.62–6394.40
**WF** (**L/person/day**)			
Estimated footprint ^1^	2702.53	2107.00–3436.06	469.01–14,597.74
Adjusted to 1000 kcal	1834.03	1536.60–2264.23	777.45–5430.45
**EF** (**m^2^/person/day**)			
Estimated footprint ^1^	20.47	15.45–27.24	4.26–102.26
Adjusted to 1000 kcal	14.29	10.83–18.50	5.78–66.19

^1^ The median of calories was 1289.08. IQR = interquartile range; CF = carbon footprint; WF = water footprint; EF = ecological footprint; gCO_2_eq = grams of CO_2_ equivalent; L = liters; m^2^ = square meters; kcal = kilocalories.

**Table 3 foods-11-03842-t003:** Socioeconomic characteristics, food purchase practices, and environmental footprints (CF, WF, and EF) per 1000 kcal of adults and elderlies living in the municipality of Natal/RN, Brazil, BRAZUCA Natal Study.

Variables	*n*	%	CI 95%	CF *		WF *		EF *	
				1000 kcal	*p*-Value	1000 kcal	*p*-Value	1000 kcal	*p*-Value
**Sex**									
Male	173	42.1	37.2–46.7	1957.29 ^a^	0.05	1915.43 ^a^	0.02	14.28 ^a^	0.85
Female	238	57.9	53.3–62.8	1828.36 ^b^		1745.90 ^b^		14.36 ^a^	
**Age group**									
Adults	220	53.5	48.7–58.9	2106.32 ^a^	0.00	1889.73 ^a^	0.01	14.82 ^a^	0.06
Elderly	191	46.5	41.1–51.3	1797.76 ^b^		1765.35 ^b^		13.71 ^a^	
**Ethnicity**									
White	145	35.3	30.7–40.1	2001.98 ^a^	0.36	1864.03 ^a^	0.28	14.33 ^a^	0.31
Non-white	266	64.7	59.9–69.3	1975.56 ^a^		1813.65 ^a^		14.20 ^a^	
**Schooling** (**years of study**) **^1^**									
0–5	146	35.8	31.1–40.4	1812.52 ^a^	0.06	1757.70 ^a^	0.21	14.45 ^a^	0.47
6–9	62	15.2	11.8–18.6	1717.28 ^a^		1731.63 ^a^		13.36 ^a^	
10–13	126	30.9	26.5–35.8	1968.77 ^a^		1835.12 ^a^		14.00 ^a^	
≥14	74	18.1	14.5–22.1	2168.33 ^a^		1951.36 ^a^		14.88 ^a^	
**Monthly per capita family income** (**BRL**) **^2^**									
<249.50 (≅USD 47.22)	84	20.9	17.0–24.9	1710.69 ^b^	0.00	1715.54 ^b^	0.00	13.60 ^a^	0.95
249.50–449.45 (≅USD 47.22–85.07)	76	19.0	15.2–22.9	1816.02 ^a,b^		1885.33 ^a,b^		13.66 ^a^	
449.46–762.67 (≅USD 85.07–144.35)	81	20.2	16.2–23.9	1900.64 ^a,b^		1771.22 ^b^		14.82 ^a^	
762.68–1559.99 (≅USD 144.36–295.08)	80	20.0	16.2–23.7	2095.09 ^a,b^		1866.13 ^a,b^		14.16 ^a^	
≥1560.00 (≅USD 295.27)	80	20.0	16.5–23.9	2186.92 ^a^		2092.83 ^a^		14.78 ^a^	
**Frequency of fast food purchase ^3,†^**									
Never	262	65.7	61.2–69.9	1797.51 ^b^	0.00	1737.21 ^b^	0.00	13.68 ^a^	0.34
Sometimes	117	29.3	25.1–33.8	2112.25 ^a^		2007.87 ^a^		14.81 ^a^	
Often	20	5.0	3.0–7.3	2761.45 ^a^		2097.04 ^a^		16.74 ^a^	
**Frequency of lunch or dinner at snack bars ^4^**									
Never	230	57.5	52.5–62.2	1813.81 ^b^	0.00	1784.65 ^b^	0.02	14.31 ^a^	0.39
Hardly ever	119	29.8	25.3–34.0	1987.81 ^b^		1853.54 ^b^		13.10 ^a^	
At least once a week	51	12.8	9.3–16.3	2347.80 ^a^		2065.91 ^a^		15.09 ^a^	
**Frequency of food purchase at street fairs ^3^**									
Never	116	29.1	34.8–33.6	2007.63 ^a^	0.41	1872.45 ^a^	0.39	13.97 ^a^	0.26
Sometimes	154	38.6	33.8–43.4	1947.70 ^a^		1831.42 ^a^		15.00 ^a^	
Often	129	32.3	28.1–36.8	1830.99 ^a^		1788.39 ^a^		13.71 ^a^	
**Frequency of use of food delivery services ^3^**									
Never	247	61.9	57.4–66.4	1825.72 ^a^	0.03	1783.62 ^a^	0.12	14.32 ^a^	0,33
Sometimes	123	30.8	26.6–35.3	2057.07 ^a,b^		1919.23 ^a^		13.93 ^a^	
Often	29	7.3	4.8–10.0	2213.07 ^b^		1828.81 ^a^		14.77 ^a^	

^1^ Three values answered as “didn’t answer/didn’t know” (DA/DK) were ignored; ^2^ ten values answered as DA/DK were ignored; ^3^ twelve values answered as DA/DK were ignored; ^4^ eleven values answered as DA/DK were ignored; ^†^ fast food was considered as hamburger, pizza, pastries, and other highly processed foods, while snack bars serve quick meals that include sandwiches, natural juices, smoothies, coffee, cakes, sweets, and other products that may or may not be considered fast food; * different letters in the columns indicate a statistical difference between values according to Mann–Whitney U or Kruskal–Wallis tests (*p* < 0.05). The same letters do not significantly differ.

## Data Availability

The data presented in this study are available on request from the corresponding author. The data are not made public due to Brazilian law (CNS Resolution 466/12), which enforces the confidentiality and privacy of participants and data during all phases of research. The environmental footprint database used in this study is openly available at http://www.livrosabertos.sibi.usp.br/portaldelivrosUSP/catalog/view/442/394/1603 (https://doi.org/10.11606/9788588848405).

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
