# Peer review of "Dietary Environmental Footprints and Their Association with Socioeconomic Factors and Food Purchase Practices: BRAZUCA Natal Study"

_foods, 2022, doi:10.3390/foods11233842_

Round 1

Reviewer 1 Report

In the paper, the author estimate the carbon (CF), water (WF), and ecological (EF) footprints of the diet of residents in the city of Natal, Brazil, and to verify their association with socioeconomic factors and food purchase practices. And, it is important to emphasize that lower footprint values do not necessarily indicate a healthier and more sustainable diet. Howver, a few issues are needed to be clarified.

1. It is suggested to add the classification of dietary differences in Metrics in Table1.

2. Check the full text. There are several grammatical errors.

3. No study was designed on the correlation between patients and ENF, CF, and WF. It is obvious that they are unable to collect food or there are few channels to collect food.

4. In Table 5, Some fonts do not conform to the specification.

Reviewer 2 Report

The manuscript describe the dietary environmental footprints and their association with socioeconomic factors and food purchase practices (BRAZUCA natal study).

1. Title - What is BRAZUCA? a city? why all capital letters?
2. Abstract - How do you identify p = 0.0? or p<0.05?
3. Line 45 - not sure whether graphical abstract should be included in the manuscript. check journal format.
4. Introduction - revise your paragraph. combination of paragraph is suggested. some of paragraph only 2-3 sentences.
5. Since authors focused their study in Brazil, it is suggested to include discussion regards to national/government policy that support study.
6. Provide overall flowchart of methodology.
7. Line 127 - why the data collected in pre-pandemic period? June 2019 not pandemic yet. pandemic starts in the end of 2019/early 2020. any justification? discuss.
8. Line 148 - All steps were guided by manuals and SOP. Explain clearly.
9. Provide references/citation for equation used in this study.
10. Cite references for method used (if necessary)
11. Line 257. Why authors chose Kolmogorov-Smirnov test, Mann-Whitney U test, Kruskal-Wallis other than statistical methods? Provide justification in manuscript - the criteria of chosen statistical test.
12. section 3.1 - how about comparison with other countries? more comparison, not only Sweden.
13. Figure 5 - make it half page or 1 page. more clear.
14. Line 565 - Some statistical analyzes could not be performed.. which part of study? Explain
15. Conclusion too long. Simplify in 1 paragraph. Provide recommendations.
16. Provide more updated references - 2022.

Reviewer 3 Report

It is a very interesting study and written in good quality. However, I have a few comments related to the characteristics of the studied group. In my opinion, the authors should prepare and add in the material and methodical part a table with the following information: age, sex, education, place of residence, place of work, financial situation, family, etc.

Interesting information may concern the health part, for example individual health assessment, BMI, physical activity. Was such research carried out in this experiment?
